# Early Diagnosis of Acute Myocarditis in the ED: Proposal of a New ECG-Based Protocol

**DOI:** 10.3390/diagnostics12020481

**Published:** 2022-02-13

**Authors:** Isabelle Piazza, Paolo Ferrero, Alessio Marra, Roberto Cosentini

**Affiliations:** 1Emergency Department, ASST Papa Giovanni XXIII, 24127 Bergamo, Italy; isabelle.isi1992@gmail.com (I.P.); a.marra@asst-pg23.it (A.M.); 2Scuola di Specializzazione in Medicina d’Emergenza e Urgenza, Università Degli Studi di Milano, 20122 Milan, Italy; 3Pediatric and Adult Congenital Heart Centre, IRCCS-Policlinico San Donato, 20097 San Donato Milanese, Italy; ferrerop41@gmail.com

**Keywords:** myocarditis, ECG, QRS fragmentation, emergency department, critical care medicine, diagnostic and therapeutic protocols in emergency medicine

## Abstract

The diagnosis of acute myocarditis (AM) is based on a multi-parametric assessment including clinical presentation, ECG, imaging and biomarkers. Fragmented QRS (fQRS) might be an additional diagnostic sign in patients with proven AM. The main objective of this study was to assess the diagnostic yield of fQRS in patients with suspected AM presenting to the emergency department (ED). Patients admitted between January 2016 and March 2021 with a proven diagnosis of AM, according to clinical, cardiac magnetic resonance (CMR) and/or histologic criteria, were included in the analysis. In total, 51 patients were analyzed (41 men, 78%), with a median age of 36 (29–45) years. Thirty-three (65%) patients had prodromal flu-like symptoms. Patients presented to the ED mostly complaining of chest pain (68%) and palpitations (21%). Seven (14%) patients experienced cardiac arrest, one of whom died. At presentation, 40 patients (78%) displayed fQRS, and 10 (20%) presented ventricular arrhythmias. All the surviving patients underwent CMR and displayed late gadolinium enhancement (LGE). ECG leads showed that fQRS matched the LGE distribution in 38 patients (95%). The presence of fQRS is a simple clinical bedside tool to support the initial suspect of AM in the emergency department and to guide the most appropriate clinical workup.

## 1. Introduction

Myocarditis is an inflammatory cardiac muscular disease with an extremely heterogeneous etiology [1,2]. The causes may be infective (e.g., bacterial or viral) or non-infective (e.g., drugs, physical and biochemical agents, systemic inflammatory diseases, autoimmune causes) [3,4]. Overall, there are various incidence estimations of myocarditis, ranging from 0.02 to 1.5%, based upon autopsy studies [5,6,7]. Despite invasive and non-invasive treatments, a 10% mortality rate at 3-year follow-up still remains [8]. The average age of patients with myocarditis ranges from 20–51 years [6,9], and it is more common in men [10,11,12].

The diagnosis of acute myocarditis (AM) is based on consistent symptoms, electrocardiogram (ECG), cardiac imaging, serum biomarkers indicative of myocardial damage and endomyocardial biopsy (EMB) [1]. EMB represents the diagnostic gold standard, but in the last years, it has been replaced by cardiac magnetic resonance (CMR) in routine clinical practice [13]. Although aspecific, ECG is still the first-line exam performed in the ED for various cardiac conditions, including AM. Indeed, ECG, together with clinical picture and increase in serum biomarkers, usually allows provisional differential diagnosis between suspected AM and other conditions such as acute coronary syndromes [13,14,15].

The fragmentation of QRS (fQRS) is a diagnostic and prognostic index described in different heart diseases sharing common histopathological characteristics, namely, infiltrates and/or myocardial fibrosis [16]. The common diagnostic feature of QRS fragmentation, described in different clinical setting, is the presence of discernible multiple notches in either the R or S component of the QRS [17]. fQRS has been extensively studied in ischemic heart disease, showing a higher sensitivity as compared to Q wave for the detection of myocardial scar. Moreover, evidences in the setting of ischemic cardiomyopathy suggest a significant correlation with ventricular function and association with cardiac events at follow-up [18,19,20].

We have previously described the reproducibility of fragmented QRS as a complementary diagnostic sign in patients with proven AM in adult and pediatric populations [21,22,23]. This observation stemmed from the hypothesis that pathological changes in myocardium of patients with AM might translate into anisotropic conduction expressed as fQRS in one or more leads.

The “early identification” of patients in the Emergency Department (ED) is the most challenging issue related to myocarditis management due to the risk of rapid evolution towards heart failure and life-threatening arrhythmias. Recently, an Italian group proposed a clinical score for a fast evaluation and identification of patients suffering from myocarditis and attending the ED, not including ECG [24].

We sought to investigate the clinical utility of a systematic search for QRS fragmentation at the time of ED presentation to support the diagnosis of acute myocarditis and to guide early management.

## 2. Materials and Methods

This was a monocentric, retrospective study involving the ED of Papa Giovanni XXIII in Bergamo, Italy, a tertiary hospital with over 90,000 visits/year. Patients admitted between January 2016 and March 2021 with a proven diagnosis of AM, according to clinical, CMR and/or histologic criteria, were retrospectively analyzed. Clinical variables recorded at baseline were: 12-leads ECG, echocardiography and laboratory findings. Hospital records of admitted patients were also retrieved in order to assess the association of initial presence vs. absence of fQRS with the ascertainment of AM diagnosis. As previously published, fQRS was defined as the presence of an additional R wave (R’) or notching in the nadir of the S wave or the presence of >1 R’ in at least 2 contiguous leads [21,22].

According to the institutional protocol, CMR was performed in all patients admitted with the suspect of AM. CMR was considered diagnostic of AM in the presence of Lake Luis criteria [25].

Distribution of late gadolinium enhancement (LGE) was categorized into three patterns according to the myocardial segment involved: antero-septal, infero-lateral and spotted.

EMB was performed in selected patients from the right side of the interventricular septum under fluoroscopic guidance. Selective coronary angiography was routinely performed in patients older than 40 year or in case of ischemic etiology suspect.

Troponin values were standardized as ratio between patient levels and upper limit of the normal range in order to overcome change of reference values over time.

### Statistical Analysis

Continuous variables were reported as median and inter-quartile ranges and compared by Wilkinson rank sum test. Normality of continuous variables was assessed by visually inspecting the distribution histograms. Categorical variables were presented as counts and percentages and compared by χ^2^ or Fisher exact test.

A *p* value of 0.05 was assumed as cut-off for statistical significance.

Analysis was performed with STATA 11.0 by Stata Corp.

## 3. Results

### 3.1. Clinical Presentation and In-Hospital Course

In total, 51 patients were analyzed (41 men, 78%), 42 (82%) with Caucasian ethnicity and with a median age of 36 (29–45) years. All patients reported prodromal symptoms: 33 (65%) patients had flu-like syndrome symptoms, 16 (31%) had pharyngodinia and 8 (16%) had gastrointestinal syndrome. The symptoms at ED presentation were chest pain (68%), palpitations (21%) and syncope (8%). Seven (14%) patients experienced cardiac arrest, one of whom died. (Table 1)

### 3.2. Diagnostic Findings and In-Hospital Course

Forty-five (88%) patients had C reactive protein (CRP) elevation, and all patients had elevated Troponin levels. The median CRP was 4.4 (2.5–13.2) mg/dL (n.v. < 1 mg/dL). The highly sensitive Troponin I ratio peak was 176.8 (13.2–438.7) (Table 2). The median ratio of Troponin was not significantly different in patients displaying fQRS, 38 (6.27–116) vs. 20.6 (3.3–52.8) *p* = 0.24.

The median ejection fraction (EF) was 55% (45–60); 23 patients (45%) had wall motion abnormalities and 9 (18%) had pericardial effusion. All 50 surviving patients underwent CMR during the acute phase (<21 days from admission). Moreover, 11 (22%) patients showed anterior/septal LGE, 21 (42%) patients showed inferior/lateral LGE and 18 (36%) showed spot LGE. Ischemic heart disease was excluded by coronary angiography in 25 (49%) patients. Biopsy was performed in 6 (12%) patients (Table 2).

A total of 35 (68%) patients were admitted to an intensive care unit (ICU), while the 16 (32%) remaining patients were admitted to a cardiology ward. Eight (15%) patients presented with cardiogenic shock or low cardiac output syndrome and needed an inotropic support early during the hospitalization. This latter group included seven (14%) patients who underwent extracorporeal membrane oxygenator (ECMO) (Table 1).

### 3.3. ECG and QRS Fragmentation

At presentation, ECG was abnormal in 48 (94%) patients. Of these, 19 (37%) ECGs displayed aspecific findings, 23 (45%) showed ST elevation and 7 (14%) showed ST depression. fQRS was visible in 40 (78%) patients. Ventricular arrhythmias occurred in 10 (20%) patients, non-sustained ventricular tachycardia (NSVT) and frequent premature contractions (PVC) occurred in 7 (14%) patients and supra-ventricular arrhythmia occurred in 2 (4%). Two (4%) patients showed first-degree atrio-ventricular block and fourteen (27%) a left or right bundle block. In eight (16%) patients, a device implantation was necessary during the hospitalization or follow-up.

There was a trend towards a higher incidence of ventricular arrhythmias in patients with fQRS (25% vs. 10%, *p* = 0.2).

Evidence of QRS fragmentation occurred with a latency of 3 (2–6) days from the onset of symptoms. ECG leads showed that fQRS matched the myocardial area of the LGE distribution in 38 patients (95%) (Table 3).

Overall fQRS was identified in 40 out of 50 patients, with CMR-documented acute myocarditis accounting for a sensitivity of 78%.

## 4. Discussion

Acute myocarditis may present with a wide clinical spectrum, ranging from mild aspecific symptoms to cardiogenic shock and sudden death [26,27,28]. However, an early high-degree of suspicion is of paramount importance, as patients’ conditions can deteriorate very quickly. Diagnosis in the ED is particularly challenging as common diagnostic tools used in the acute setting, and ECG in particular, are commonly believed to be aspecific.

The belief in the lack of specificity of the ECG has been recently challenged by recent data suggesting that fQRS might be a new additional specific sign correlating with the presence and topographic distribution of myocardial inflammation (Figure 1) [21,22].

In this study, we observed that systematic search of fQRS in the first ECG recorded in the ED may be useful for the early triage of patients with suspected myocarditis (Figure 2).

It has previously shown that the presence of QRS fragmentation is associated with evidence of LGE at CMR and impairment of contractility [29,30]. Based on this background, we can hypothesize that the early recognition of myocardial inflammation indicated by presence and extension of fQRS might identify a subset of higher-risk patients, which may deserve early specific work-up (Figure 2). In our cohort, fQRS initially correctly classified 40 out of 51 patients in which acute myocarditis was eventually ascertained. Interestingly, QRS fragmentation appeared with a latency of a few days after symptoms onset. This is consistent with the time elapsed before a critical threshold of inflammation is reached in order to be detected on surface ECG.

We can also speculate that the presence of fQRS, being associated with the extension of inflammation and scar, may be considered a marker of arrhythmic risk, prompting ECG monitoring during ED observation [31]. Indeed, in our cohort, we observed a tendential association of fQRS with a higher incidence of ventricular tachycardia.

If confirmed on larger samples that this preliminary observation might support the clinical validation of a new triage of patients with suspected myocarditis based on the early specific search for fQRS on the ECG at presentation.

### Limitations

This study has several limitations. Firstly, this is a retrospective, single-center study involving a limited sample size. Furthermore, fQRS may be particularly subtle, and some training is needed in order to accurately and reproducibly identify this sign. This may have inflated the sensitivity of fQRS in our cohort. Furthermore, the presence of fQRS was assessed by a single investigator, therefore, we do not have data about inter-observer agreement.

Given the variability of the myocardial inflammation evolution, the sensitivity and specificity of this sign may largely vary according to the time elapsed between symptoms onset and ECG recording.

## 5. Conclusions

ECG is strongly recommended in patients with suspected AM presenting with typical or atypical chest pain or rhythm disturbances. In addition to common aspecific ECG changes, the presence of fQRS is a simple clinical tool to corroborate the initial suspect of AM in the emergency department and to guide the following clinical workup.

## Figures and Tables

**Figure 1 diagnostics-12-00481-f001:**
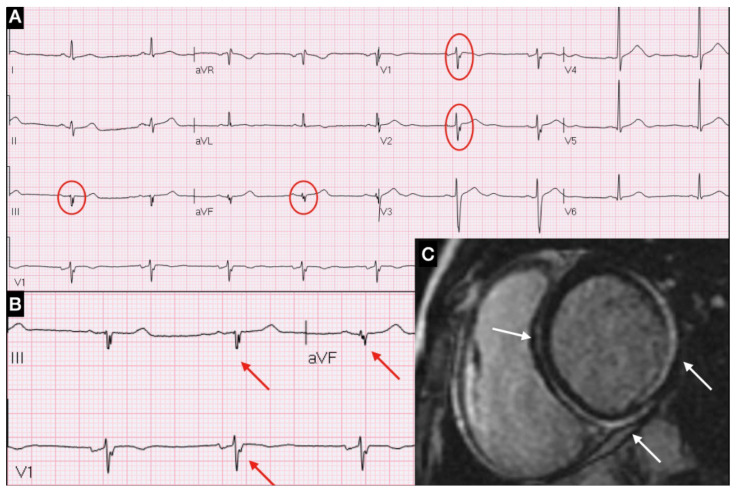
(**A**) ECG showing fQRA in leads V1, V2, DIII and aVF; (**B**) Magnification of fQRS; (**C**) CMR short-axis view with LGE in inferior-lateral and septal areas.

**Figure 2 diagnostics-12-00481-f002:**
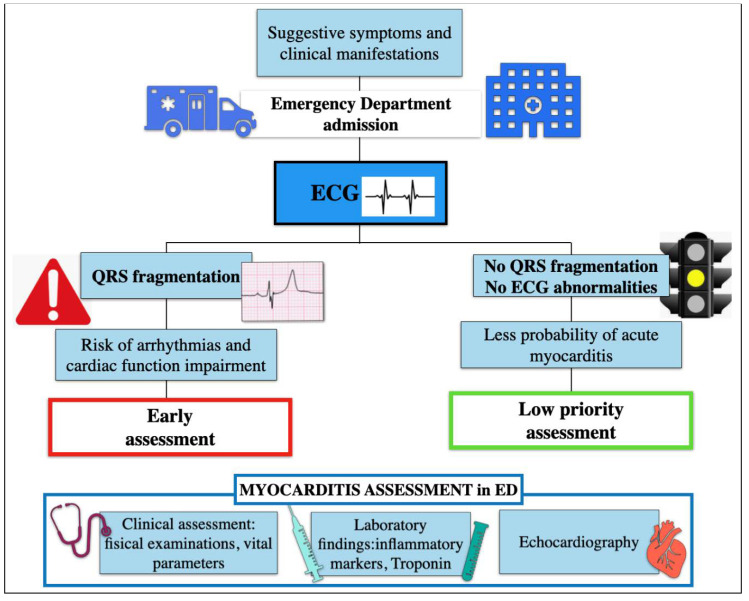
Algorithm assessment of patient with suspected myocarditis in Emergency Department.

**Table 1 diagnostics-12-00481-t001:** General demographics and clinical and diagnostic characteristics of the population at presentation at the emergency department. IQR: interquartile range.

	*n* = 51
Age (years), median (IQR)	36 (29–45)
Male, *n* (%)	41 (78)
Caucasian ethnicity	42 (82)
Prodromal Symptoms, *n* (%)	
Prodromal symptoms duration, median (IQR)	3 (2–6.5)
Fever/Flu-like syndrome, *n* (%)	33 (65)
Pharyngeal pain, *n* (%)	16 (31)
Gastrointestinal disorders, *n* (%)	8 (16)
Clinical presentation/In-hospital course	
Chest pain, *n* (%)	35 (68)
Palpitations, *n* (%)	11 (21)
Syncope, *n* (%)	4 (8)
Cardiac Arrest, *n* (%)	7 (14)
Inotropic support, *n* (%)	8 (15)
Mechanical respiratory/circulatory support, *n* (%)	7 (14)

**Table 2 diagnostics-12-00481-t002:** Diagnostic findings in patients admitted to ED with AMAM: acute myocarditis; CRP: C reactive protein; LVEF: left ventricular ejection fraction; CMR: cardiac magnetic resonance.

	*n* = 51
Laboratory findings at presentation	
CRP mg/dL, median (IQR)	4.4 (2.5–13.2)
Troponin ratio, median (IQR)	176.8 (13.2–438.7)
Echocardiography, *n* (%)	
LVEF (%), median (IQR)	55 (45–60)
Wall motion abnormalities, *n* (%)	23 (45)
Pericardial effusion, *n* (%)	9 (18)
Biopsy, *n* (%)	6 (12)
Coronary angiography, *n* (%)	25 (49)
CMR, *n* (%)	*n* = 50
LVEF (%), median (%)	58 (52–61.2)
LGE, *n* (%)	50 (100)
Antero-septal LGE, *n* (%)	11 (22)
Infero-lateral LGE, *n* (%)	21 (42)
Other-pattern LGE, *n* (%)	18 (36)

**Table 3 diagnostics-12-00481-t003:** ECG and rhythm disturbance in patients admitted with AM.

	*n* = 51
ST elevation, *n* (%)	23 (45)
Aspecific abnormalities, *n* (%)	19 (37)
ST depression, *n* (%)	7 (14)
Ventricular Arrhythmia, *n* (%)	10 (20)
NSVT or frequent PVC, *n* (%)	7 (14)
Supraventricular arrhythmia, *n* (%)	2 (4)
Device implantation, *n* (%)	8 (16)
QRS fragmentation, *n* (%)	40 (78)
Latency of fQRS occurrence (days), median (IQR)	3 (2–6)
fQRS matching LGE distribution, *n* (%)	38 (95)
Wall motion anomalies, *n* (%)	23 (45)
Pericardial effusion, *n* (%)	9 (18)

AM: acute myocarditis; NSVT: non sustained ventricular tachicardia; PVC: premature ventricular contractions; fQRS: fragmented QRS; LGE: late gadolinium enhancement.

## Data Availability

The data presented in this study are available on request from the corresponding author.

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
