# Peer review of "Early Diagnosis of Acute Myocarditis in the ED: Proposal of a New ECG-Based Protocol"

_diagnostics, 2022, doi:10.3390/diagnostics12020481_

Round 1

Reviewer 1 Report

The article "Early Diagnosis of Acute Myocarditis in the ED: Proposal of e 2 New ECG Based Protocol" has been presented in an orderly mode and the limitations are appropriate to the type of research performed. An excelllent atempt for the early differential diagnosis of acute myocarditis by fQRS in the the patient arriving in the emergency department. The fQRS together with CMR (has sensitivity value 75%-100% and specificity of 90%) would be usful for such path. 

Author Response

We deeply thank the reviewer for appreciating our attempt to optimize ECG use to achieve early diagnosis of myocarditis in the ED. 

Reviewer 2 Report

Statistical Analysis deserves its own paragraph within Methods.

The following report should be included:

Significance of a Fragmented QRS Complex Versus a Q Wave in Patients With Coronary Artery Disease

Mithilesh K. Das, Bilal Khan, Sony Jacob, Awaneesh Kumar, and Jo Mahenthiran

Originally published 22 May 2006

https://doi.org/10.1161/CIRCULATIONAHA.105.595892

Circulation. 2006;113:2495–2501

Limitations of the study should be acknowledged, including sources of potential bias.

Author Response

  1. Statistical Analysis deserves its own paragraph within Methods.

We added a paragraph for statistical analysis as suggested.

  1. The following report should be included: Significance of a Fragmented QRS Complex Versus a Q Wave in Patients With Coronary Artery Disease. Mithilesh K. Das, Bilal Khan, Sony Jacob, Awaneesh Kumar, and Jo Mahenthiran Originally published 22 May 2006 https://doi.org/10.1161/CIRCULATIONAHA.105.595892. Circulation. 2006;113:2495–2501

We thank the reviewer for this comment. We have added a sentence about comparison between fQRS and Q wave in coronary artery disease together with the relative reference.

  1. Limitations of the study should be acknowledged, including sources of potential bias.

We have expanded the limitation section, including more detailed mention of possible bias.

  1. Moderate English changes required 

Paper style and language have been reviewed. 

Round 2

Reviewer 2 Report

-